# A Survey on Undergraduate Medical Students’ Perception of COVID-19 Vaccination

**DOI:** 10.3390/vaccines10091464

**Published:** 2022-09-03

**Authors:** Rosagemma Ciliberti, Francesca Lantieri, Rosario Barranco, Camilla Tettamanti, Alessandro Bonsignore, Francesco Ventura

**Affiliations:** 1Section History of Medicine and Bioethics, Health Science Department (DISSAL), University of Genoa, Via De Toni 12, 16132 Genova, Italy; 2Biostatistics Unit, Health Science Department (DISSAL), University of Genoa, Via Pastore 1, 16132 Genova, Italy; 3Section of Legal Medicine, Health Science Department (DISSAL), University of Genoa, Via De Toni 12, 16132 Genova, Italy; 4IRCCS—Ospedale Policlinico San Martino Teaching Hospital, Largo R. Benzi 10, 16132 Genova, Italy

**Keywords:** COVID-19 vaccine, medical students, vaccine hesitancy, vaccination acceptance, vaccine efficacy, green pass

## Abstract

The objectives of this study were to obtain information on medical students’ attitudes toward COVID-19 vaccination and to identify the main barriers to its acceptance. We conducted an anonymous online survey on a sample of undergraduate medical students from one main Italian University. The questions were aimed at exploring their attitudes toward vaccination to prevent COVID-19, their perceptions of the risk/threat of COVID-19 and the factors associated with their attitudes toward COVID-19 vaccination. A high percentage of students in our sample stated that they had been vaccinated or that they intended to be vaccinated against the COVID-19 coronavirus. A total of 239 questionnaires were analyzed. Age, social, geographic and demographic characteristics, health conditions and interest in vaccination were recorded; 93% of the students declared that they encouraged vaccination and 83% stated that the reason was “Moral responsibility towards the community”. Four students had not yet been vaccinated, mainly because of “Contradictory information on efficacy and safety”. The Likert-type questions revealed high agreement on the importance of vaccination and whether it should be made mandatory (“indispensable tool” and “ethical duty” were cited to explain this position). The results show a high level of acceptance of COVID-19 vaccination among these medical undergraduates who, being halfway through their training and involved in clinical practice, are already in possession of specific scientific knowledge and, to a small extent, come from different areas of Italy.

## 1. Introduction

Ten years ago, Rosoff et al. [1] published their concern regarding equality in a pandemic context, in that vulnerable and socially marginalized groups risk being excluded from health programs Globally, as of 1 February 2022, 376,478,335 confirmed cases of COVID-19, and 5,666,064 deaths, have been reported to the World Health Organization (WHO) [2].

Italy was the first European country to be hit by COVID-19 in the early stages of the pandemic, with >400,000 confirmed cases and >36,000 COVID-attributed deaths being recorded as of mid-October 2020, and was the first to declare lockdown in March 2020 [2,3].

Vaccination campaigns against COVID-19 are progressively allowing severe restrictions on social, economic, cultural and educational activities to be relaxed. 

As of 4 February 2022, roughly 47.4 million people in Italy had completed the regular course of vaccination against COVID-19, corresponding to 87.9% of the total population over 12 years of age [4].

According to UNESCO, as of 8 April 2020, more than 188 countries had implemented nationwide school and university closures, impacting on over 90% of the world’s student population.

University students are at high risk of COVID-19 infection and/or transmitting the infection to vulnerable people outside the university setting, owing to many factors: prolonged close contact among high numbers of students during regular face-to-face teaching, the use of public transport in traveling to and from university, frequent contacts among students during their everyday lives, and the high levels of mixing in halls of residence [5]. Moreover, the involvement of medical students, during their clinical training, in providing care to patients together with the high transmissibility of the COVID -19 infection, further increase the risk in this subpopulation of contracting and transmitting the infection, especially to vulnerable people such as those with cardiovascular disease, cancer or suffering from chronic diseases [6,7]. In addition, in an emergency, marked by the difficulty of communicating with health professionals, medical students are often called on for health advice from family and friends. Nonetheless, medical students often have less knowledge and awareness than medical professionals about the risks of infections that may be contracted and transmitted to others, including patients in hospitals and medical centers, because of their still limited professional experience. In addition, university students are increasingly being recognized as a vulnerable population, and they are experiencing higher levels of anxiety, depression, and stress than the general population during the COVID-19 pandemic [8]. The university period is a critical one, as it marks the student’s passage to independent adult life. Moreover, the pressure of university life and the need to achieve good academic results can place a heavy burden on the student’s mental health and may lead to long-term consequences [9]. In this regard, studies conducted among university students in France, Spain and Poland during the COVID-19 lockdown showed an increase in stress and anxiety in about 60% of subjects [10,11].

To date, although great efforts to cope with this crisis have been made, no efficacious specific treatment for COVID-19 is yet available [12,13].

Implementing safe and effective vaccination campaigns is an important means of fighting the pandemic, in order to return to “normality” [14]. Several vaccines have been approved for use against coronavirus disease (COVID-19) and have been distributed throughout the world, offering the best opportunity to exit from the COVID-19 crisis [15].

In Italy, as well as in Europe, the following vaccines have been authorized and administered: Pfizer—BioNTech (approved on 21 December 2020), mRNA-1273 Moderna (authorized on 6 January 2021); ChAdOx1 nCov-19 AstraZeneca (authorized 29 January 2021); COVID-19 Vaccine Janssen, Johnson & Johnson (authorized 11 March 2021) [16].

The WHO reported that, as of 31 January 2022, a total of 9,901,135,980 vaccine doses had been administered [2].

Although vaccination is an essential strategy against the transmission of the COVID-19 virus, the availability of safe and effective vaccines is not, in itself, sufficient in order to protect vulnerable groups (those with serious pathologies such as cardiovascular disease or cancer). Achieving herd immunity is an important means of reducing hospitalizations and severe disease, and it is the main weapon in the fight to overcome pandemics. Vaccine safety and adherence to vaccination are fundamental factors in the success or failure of a vaccination program [17]. Indeed, the WHO considers vaccine hesitancy to be among the top ten threats to global health [18]. 

University students, particularly those in the healthcare disciplines, constitute a category of particular interest, as their future professional role will enable them to inform and educate citizens regarding vaccination, which is of great ethical and social importance. Indeed, the positive opinion of university students can have a significant influence on social trends [19]. In addition, age seems to play a key role in determining vaccine acceptance, though its effects are reported to vary in different countries [20].

This study aims to assess vaccination coverage and to investigate COVID-19 behaviors and attitudes toward vaccination among university students of medicine in Italy, to identify the factors that influence students’ decisions regarding vaccination.

## 2. Materials and Methods

### 2.1. Questionnaire

A sample of undergraduate medical students from the University of Genoa was recruited in December 2021; at that time, the third round of vaccination had started in Italy and vaccination had been made obligatory for health professionals (but not students) and law enforcement officers. During the study period, Italian universities were providing tuition via the Internet, owing to the control measures imposed during the fourth wave of the COVID-19 pandemic. An online, anonymous questionnaire was proposed to the students who were enrolled in the third year of the School of Medicine, on the occasion of a core lecture; in Italy, such students are typically between 20 and 22 years old. Written in the Italian language, the questionnaire was administered via the e-learning platform of Genoa University (AulaWeb), which is based on Moodle. The questionnaire contained 21 questions (two of which were alternative, depending on the previous answer, for a total of 20 questions for each participant) and consisted of two sections. In the first section, participants’ sociodemographic data (age, sex, nationality, region of residence at the beginning of the pandemic, domestic cohabitation) were collected. In addition, students were asked whether they had been infected by SARS-Cov-2 disease, whether they suffered from any chronic diseases, and whether they had already been vaccinated. In the second section, participants were questioned about vaccination against COVID-19. Specifically, they were asked about their reasons for being/not being vaccinated, their perceptions of experimentation for the approved vaccines, their attitudes toward vaccination, and their opinion on whether vaccination should be compulsory.

Eleven questions had closed answers, with the additional option to tick “I prefer not to answer”. In two cases, more than one answer was possible (reasons for being or, alternatively, for not being vaccinated and topics on which the participants were most interested in getting further information).

The 9 questions investigating the student’s attitude toward vaccination and its compulsory imposition were 11-point Likert-type agreement items: from 0 (totally disagree) to 10 (totally agree). These latter questions, in particular, investigated—on a rating scale—whether the students considered the vaccination crucial and ethical in order to protect individual and public health (two questions); their opinion about compulsory vaccination for general population and University environments (3 questions) and alternative strategies (caution and distancing and out-of-pocket costs, without burdening the public health system, for people with COVID-19 who had refused vaccination, 2 questions); and lastly, whether they considered the vaccine had been adequately tested or—on the contrary—tested hastily and therefore with uncertain efficacy and safety (2 questions). 

The questionnaire was designed and elaborated, and then discussed and reviewed, by a team of experts on ethics, forensic and biostatistics aspects. The team was also aware of the training type and level of the students enrolled and discussed the acceptability and comprehensibility of the questions. The reliability was evaluated by internal consistency with Cronbach’s Alpha.

The questionnaire, translated in English, is downloadable on line as Appendix A.

### 2.2. Statistical Methods

Categorical data are reported as counts and percentages. The percentages were calculated after excluding unanswered questions (“I prefer not to answer”). Likert-type answers are reported as medians and frequencies of answers for each score. Regions where the students were living when the pandemic began (March 2020) were also categorized as: most severely affected (north-eastern Italian regions: Lombardy, Trentino Alto Adige, Veneto, N = 35 respondents), less severely affected (north-western regions: Liguria, Piedmont, Valle D’Aosta, Toscana, N = 188 respondents) and regions where COVID circulated least (central and southern regions: Abruzzo, Basilicata, Calabria, Campania, Puglia, N = 8 respondents).

Comparisons between groups (female and males, affected/not affected by chronic diseases) for categorical variables were made by means of chi-square or Fisher’s exact test. Comparison between categorical data and the geographical area of residence at the beginning of the pandemic was made by means of the Cochran-Armitage test. The Likert items were compared between genders and between geographical areas by means of the non-parametric Mann-Whitney and Kruskal-Wallis tests. Tests were two-tailed and significance was set at alpha = 0.05.

## 3. Results

A total of 240 university students participated in the survey, i.e., all those attending the lecture out of the 284 enrolled in the third year of the School of Medicine. Only one questionnaire was excluded from the analysis, as it had almost no answers. After the exclusion of this questionnaire, the rate of unanswered questions (among those that provided the option of not answering) was very low, ranging from 1.7–2.5% for the demographic and health questions; no respondent ticked 0 to all the Likert questions. The students’ mean age was 22.3 years (range 19–35). Table 1 shows the participants’ sociodemographic characteristics. Most respondents were female (70% of those who answered the question on gender) and 96.7% were Italian. Regarding their housing modality, among those who responded to this question, 94.4% of students lived with others (85.8% with family members, 4.7% with friends and 3.9% with a partner). A small proportion of participants (14.9% of those who answered this question) had been infected with COVID-19, while 5.9% reported a chronic pathology. Almost all respondents (98.3%) had been vaccinated, and one further student was already planning to be vaccinated. Only three students answered that they were in doubt as to whether they would be vaccinated, while none answered that they did not intend to be vaccinated.

Accordingly, most students (215, 90.0%) declared that they encouraged vaccination, with another seven students encouraging vaccination only for persons at risk of a severe disease outcome (2.9%).

Most respondents (168 out of 239) selected more than one reason for being vaccinated. 

A high percentage of the 235 respondents who had been vaccinated selected “Moral responsibility towards the community” (83.0%, Table 2); in 49 cases, this was the only reason selected. Other reasons were: “To protect my family” (69.8%) and “To protect myself” (68.1%)”, while “To get the green pass” was selected by only 24.3%; this response option was selected as the only reason by only three students, all of whom were foreigners. A higher percentage of males (34.8%) than females (20.4%) cited getting the green pass as the reason for being vaccinated (*p* = 0.0294). This motivation was selected by 25.8% of those without chronic diseases and by none of the 14 with chronic diseases (*p* = 0.0247). Of the 14 participants with chronic diseases, 13 declared their gender; these were all females. However, females indicated the green pass as a reason for vaccination less often than males, even after the 13 with a chronic disease had been excluded. Finally, eight students (3.4%) cited reasons other than those suggested in the questionnaire, five of them mentioned a sense of obligation, while two expressed reasons concerning a return to normality and freedom.

The four students who had not yet been vaccinated stated that their hesitancy was mainly due to concerns regarding “Contradictory information on efficacy and safety”, while three of the four also selected “Insufficient and hasty experimentation” and “Sense of compulsion to vaccinate”, and two expressed the belief that “the vaccination campaign is economic or other propaganda”, and one also cited “Ineffectiveness of the vaccine”. None of the four selected “fear of vaccination”.

A high percentage of respondents (190 out of 239, 79.5%, Table 3) expressed a desire to receive more information on one or more topics, with “side effects and safety” selected by 124 students (51.9%), “efficacy” selected by 122 (51.0%), and information on the different kinds of vaccines selected by 66 (27.6%). Females expressed a desire for more information slightly more frequently than males; this mainly concerned the topic of collateral effects, selected by 55.8% of females and 42.9% of males, although the difference did not reach statistical significance (*p* = 0.0942).

The Likert-type questions displayed high agreement on the importance of vaccination and whether it should be made mandatory, without any difference between females and males. Specifically, there was considerable agreement with the claims that vaccines had been properly tested, that vaccination was an “indispensable tool” and an “ethical duty”, that it should be made obligatory for the whole community” and for “all students”, and even that “those who refuse vaccination against COVID-19 should be excluded from university”. Medians ranged from 8 to 10, and between 74.9% and 93.7% of the students assigned an agreement score of 7 or more, while only between 1.7% and 13.4% assigned scores below 4. Accordingly, agreement with the statement that “Evidence of efficacy and safety is uncertain, owing to rapid development” was very low (median 2). Among both females and males, agreement with the assertions that “distancing and caution are preferable to vaccination” and “those who refuse vaccine and fall ill should bear the financial burden of treatment” was much more controversial (median 5 for both), given that the National Health Service in Italy is universal and free of charge.

The internal consistency was adequate, with a Cronbach’s Alpha of 0.777 (0.818 for the 9 Likert-type questions only), after proper reverse scoring where necessary. 

Finally, we investigated the possible association between students’ opinions and their region of residence when the pandemic began, i.e., regions most severely hit by the first wave (north-eastern Italian regions), regions less severely affected (north-west regions) and regions where SARS-Cov-2 circulated least (central and southern regions, although only eight students lived there in March 2020). No statistically significant differences emerged concerning the reasons for vaccination or other categorical items. Only interest in the type of vaccines differed, being higher in the center-south and reducing in the north-west and north-east (from 50.0 to 29.3 and 17.1% respectively, *p* = 0.0491). By contrast, most of the Likert items displayed a rising trend in agreement from the center-south to the north-west and the north-east (but decreasing regarding the uncertainty of vaccines). Specifically (Table 4), the answers regarding whether vaccines had been “properly tested”, and whether vaccination was “an ethical duty” and “should be obligatory for the whole community” and “for all students” differed significantly among the three geographical areas. As expected, gender was not significant when included in the model. Accordingly, students from north-eastern regions more often declared that they encouraged vaccination than those from north-western and central-southern regions (97.1, 91 and 50%, respectively, *p* = 0.0017).

## 4. Discussion

Our sample comprised a high percentage of students who had been vaccinated or who reported their intention to be vaccinated against the COVID-19 coronavirus. In our study, the proportion (98.3%) of vaccinated respondents was higher than the rates reported by Reno et al. (68.9%) [21] in a study of young adults conducted on the general population resident in an Italian region (Emilia-Romagna); by Del Riccio et al. (81.9%) [22], who enrolled members of the general population resident in Italy; and by the willingness toward vaccination reported by Barello et al. (86.1%) [23], who surveyed a population of Italian university students, but prior to the national vaccination campaign.

Given the greater literacy of our sample with regard to health-related issues, our findings are in line with our expectations, with the findings of Gallè et al. [24], who enrolled undergraduate students from Central and Southern Italy (91.9%), and with international data on undergraduate medical students [24,25].

In a recent survey of Japanese medical students, 89.1% had received the second dose of COVID-19 vaccine, and 90.7% stated that they would be willing to receive the vaccine in the future. Additionally, 84.5% of all participants were willing to accept a third dose of the vaccine [25]. This marked adherence to the vaccination campaign is consistent with the results of our study.

Despite the tragic health, economic and social effects of the pandemic worldwide, not everyone is willing to be vaccinated (a recent systematic review of surveys revealed an overall average COVID-19 vaccine acceptance rate of 72.5%) [26]. Vaccination hesitancy deserves particular attention in Italy, where growing mistrust of vaccines and the subsequent decline in vaccination coverage rates have forced the Italian Parliament to increase the number of mandatory vaccinations for infants [27]. Furthermore, the composition of some of these vaccines may have contributed to increasing ethical and moral doubts and, therefore, vaccine hesitancy. Indeed, in the production and/or testing process, the use of cell lines from tissue obtained from direct abortions or the presence of excipients of swine origin constitutes a major problem for some religions, which consider these practices illicit [28]. Consequently, ascertaining the factors that contribute to increasing COVID-19 vaccination coverage is essential [29].

Vaccinating medical students is a key measure in the prevention of healthcare-associated infections, owing to the close contact of these subjects with infected individuals. Our data are also in line with the findings of a large international study, which demonstrated the role of age and educational level in the acceptance of vaccination [20].

However, analysis of the answers to our questionnaire suggests that attitudes to vaccination are influenced not only by the level of health knowledge, but by other psychological factors and ethical reasons, such as moral responsibility towards the community. These data reflect not only those from international articles on the COVID-19 pandemic [23,30], but also those from the pre-COVID era [31].

It is particularly interesting that “moral responsibility towards the community” was cited by our respondents as the reason for adhering to vaccination, more than protection of their family. In accordance with the study by Barello et al. [23], these data suggest the need to promote a variety of strategies aimed at raising students’ awareness of the key role of individuals’ commitment to safeguarding their own and others’ health through vaccination. The small number of students who chose the response option “to protect myself” may be linked to a sense of invulnerability on the part of young people [32]; however, it may also stem from the awareness that elderly and immunocompromised persons are at higher risk of developing severe COVID-19 disease [33].

The motivation to be vaccinated in order to obtain the “green pass”, officially known in the European Union as the EU Digital COVID Certificate (EUDCC), deserves particular consideration. This certification, which has also been adopted by other countries [34,35], attests that the subject has been vaccinated against COVID-19 or has tested negative to the SARS-CoV-2 virus [36,37]; thus, it allows the holder to engage in a very wide variety of work activities and to access a growing list of places, gatherings, means of transport and cultural and sports events. However, persons who are exempted from the vaccination campaign because of medical conditions that have been certified in accordance with the criteria defined by the Ministry of Health are not required to possess a green pass.

The high level of agreement regarding the statements that COVID-19 vaccines have been adequately tested and that they have an essential role may be related to the type of population interviewed and the scientific knowledge in their possession. It is interesting that, in choosing the reasons for not being vaccinated, none of the four respondents who had not yet been vaccinated indicated fear of the vaccine. This finding may also be linked to the typology of the interviewees and to the knowledge acquired by them.

The high degree of agreement on the mandatory nature of vaccination for all university students and on the provision of a ban on access to university for teachers and/or students who, while not presenting medical-health impediments, refuse vaccination, deserves attention. This result differs from that of the study of older adults from southern Italy by Gallé et al. [36], which found a significantly higher percentage of people against mandatory vaccination among vaccinated individuals who took part in the study after the adoption of the green pass measure.

In order to broaden obligatory vaccination, the Italian parliament has adopted some specific measures, such as restricting admission to infant school to vaccinated children.

In accordance with this orientation, it has also been proposed that individuals who refuse anti-COVID vaccination for reasons of prejudice or ideological hostility should be called upon to take responsibility for their choice and to pay their own healthcare expenses. From the medical standpoint, however, this hypothesis introduces a complex and delicate judgment of scientific “certainty” between the behavior of the patient and the onset of the disease. From the ethical point of view, the problem is related to the need to foster social cohabitation by providing continuous care for minorities. That our respondents were substantially split on this question, however, seems to confirm the need to draw up broader and more complex strategies.

Protests against the obligation to hold a green pass in order to gain access to university have been spreading throughout the country, with students claiming the measure is discriminatory.

The Italian National Bioethics Committee (NBC), recognizing the importance of this measure, has underlined the emerging discrimination between those who have had the opportunity to be vaccinated and those who have not. It has also highlighted the problems related to the cost of serological tests and swab tests. In addition, the Committee has pointed out the psychological risk of arousing a false sense of security, and the possibility that the green pass would constitute the premise for broader measures, such as the “biological passport” or other forms of permanent surveillance of the population. Respect for human dignity and human rights is essential in order to ensure that emergency situations do not increase existing vulnerabilities and discrimination.

The present study has some limitations. First, the students’ knowledge of COVID-19 and COVID-19 vaccination could not be investigated in depth, as this would have required an excessively long questionnaire. Thus, important information regarding uncollected variables may have remained hidden. Second, the intrinsically descriptive character of the questionnaire and the dichotomous nature of several responses did not allow deeper inferential and multivariate analysis, besides investigating the gender and the geographical area were the responders lived during the first pandemic burst. In addition, the questionnaire has been extensively discussed but not tested before administration. However, no issues have emerged afterwards and the internal consistency was quite good. Finally, our sample only comprised a specific population group of undergraduate medical students, attending a single university located in northern Italy, already involved in clinical training and thus halfway toward beginning their health care profession (so being in possession of specific scientific knowledge). Accordingly, these students are not representative of the whole population of Italian medical students.

Nonetheless, these results may reflect the efficacy of teaching students about the importance and effectiveness of vaccination strategies (in contexts of diseases other than COVID) during their degree course in medicine. The same can also be said with regard to the teaching of ethics and moral responsibility.

In the awareness that mandatory measures must be accompanied by initiatives aimed at increasing adherence to vaccination, the present study demonstrates that the students surveyed are convinced of the importance of vaccination.

The data obtained from the present study may be useful in order to improve COVID-19 control strategies, with particular attention to individual respect, self-determination, and the correct messages that the world of healthcare must convey.

## Figures and Tables

**Table 1 vaccines-10-01464-t001:** Socio-demographic features and health status.

		Frequency	Percent	Percent Excluding Missing Answers
**Gender**	Female	163	68.2	70.0
	Male	70	29.3	30.0
	No answer	6	2.5	-
**Nationality**	Italians	231	96.7	96.7
	Non-Italians	8	3.3	3.3
**Housing modality**	Friends	11	4.6	4.7
	Partner	9	3.8	3.9
	Family	200	83.7	85.8
	Alone	13	5.4	5.6
	No answer	6	2.5	-
**Already infected by SARS-CoV-2**	Yes, mild/moderate	35	14.6	14.9
	Yes, severe	0	0.0	0.0
	No	200	83.7	85.1
	No answer	4	1.7	-
**Chronic disease**	No	225	94.1	94.1
	Yes	14	5.9	5.9
**COVID-19 Vaccinated**	Yes	235	98.3	98.3
	No but planning	1	0.4	0.4
	In doubt	3	1.3	1.3
	No	0	0.0	0.0

**Table 2 vaccines-10-01464-t002:** Reasons why the students were vaccinated.

	N = 235		Females (N = 162)		Males (N = 69)		
	Frequency	Percent	Frequency	Percent	Frequency	Percent	*p*-Value
**To protect myself**	160	68.1	113	69.8	46	66.7	n.s.
**To protect my family**	164	69.8	117	72.2	45	65.2	n.s.
**Moral responsibility**	195	83.0	136	84.0	57	82.6	n.s.
**Green pass**	57	24.3	33	20.4	24	34.8	0.0259
**Other reasons**	8	3.4	6	3.7	2	2.9	n.s.

**Table 3 vaccines-10-01464-t003:** Interest in further information on COVID-19 vaccination.

			Females		Males	
	Frequency	Percent	Frequency	Percent	Frequency	Percent
**Different types**	66	27.6	43	26.4	20	28.6
**Efficacy**	122	51.0	83	50.9	36	51.4
**Side effects**	124	51.9	91	55.8	30	42.9
**Other topics**	8	3.3	5	3.1	2	2.9
**Any topic in general**	190	78.2	133	79.6	53	75.7

**Table 4 vaccines-10-01464-t004:** Degree of agreement with statements on importance of vaccination and attitude toward mandatory vaccination, by geographical area.

						Comparisons *p*-Value	
	Geographical Area	N	Mean	Median	Percentage of Scores above 6	Excluding Abroad/Not Specified	North-West vs. North-East
**Properly tested**	Abroad/not specified	7	6.4	7.0	62.5		
	South/center of Italy	8	8.0	8.0	75.0	*0.0256*	
	Northwest of Italy	189	8.0	8.0	83.5		*0.0068*
	Northeast of Italy	35	9.0	9.0	100.0		
**Indispensable**	Abroad/not specified	7	6.9	7.0	62.5		
	South/center of Italy	8	9.0	10.0	87.5	0.1110	
	Northwest of Italy	189	9.3	10.0	94.2		0.0505
	Northeast of Italy	35	9.8	10.0	100.0		
**Ethical duty**	Abroad/not specified	7	6.6	9.0	62.5		
	South/center of Italy	8	9.0	10.0	87.5	*0.0092*	
	Northwest of Italy	189	9.1	10.0	93.1		*0.0024*
	Northeast of Italy	35	9.9	10.0	100.0		
**Fundamental mandatory vaccination for all**	Abroad/not specified	7	4.9	5.0	50.0		
	South/center of Italy	8	6.5	5.5	37.5	*0.0015*	
	Northwest of Italy	189	8.1	9.0	85.1		*0.0071*
	Northeast of Italy	35	9.2	10.0	100.0		
**Fundamental mandatory vaccination for students**	Abroad/not specified	7	3.4	3.0	25.0		
	South/center of Italy	8	6.7	6.5	50.0	*0.0460*	
	Northwest of Italy	189	8.5	9.0	86.2		0.1559
	Northeast of Italy	35	9.2	10.0	94.3		
**Exclude non-vaccinated from access to university**	Abroad/not specified	7	3.7	3.0	37.5		
	South/center of Italy	8	6.7	7.5	50.0	0.1984	
	Northwest of Italy	189	7.6	9.0	75.0		0.0980
	Northeast of Italy	35	8.7	9.0	88.6		
**Distancing is preferable**	Abroad/not specified	7	6.6	8.0	50.0		
	South/center of Italy	8	5.5	4.5	37.5	0.6998	
	Northwest of Italy	189	4.8	5.0	36.7		0.5274
	Northeast of Italy	35	4.5	5.0	31.4		
**Uncertain efficacy and safety**	Abroad/not specified	7	3.3	4.0	12.5		
	South/center of Italy	8	3.2	2.5	12.5	0.1261	
	Northwest of Italy	189	2.7	2.0	12.2		*0.0477*
	Northeast of Italy	35	1.9	1.0	5.7		
**Financial burden should be borne by non-vaccinated**	Abroad/not specified	7	3.4	3.0	25.0		
	South/center of Italy	8	5.6	6.5	50.0	0.8410	
	Northwest of Italy	189	5.1	5.0	41.0		0.6262
	Northeast of Italy	35	5.5	5.0	40.0		

Statistically significant *p*-values are in italics.

## Data Availability

Not applicable.

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
