# Peer review of "A Survey on Undergraduate Medical Students’ Perception of COVID-19 Vaccination"

_vaccines, 2022, doi:10.3390/vaccines10091464_

Round 1

Reviewer 1 Report (Previous Reviewer 2)

There is no conflict of interest.
The manuscript is much more clear after the corrections stating that the study was performed at the 3rd year school of a single university in Italy. It is interesting that students family place of when the pandemic began, affects their attitute towards covid i.e. the regions most severely hit by the first wave (north-eastern Italian regions), regions less severely affected (north-west regions) and regions where Sars-Cov2 circulated least (central and southern regions, alt hough only eight students lived there in March 2020).
I think that the manuscript is suitable for publication in its current formatt

Author Response

Thank you very much for your appreciation

Reviewer 2 Report (New Reviewer)

Dear Authors, 

The manuscript proposed is well written and very interesting. 

The edited part are fine and the meaning is clearer exposed.

I have just few suggestion in order to improve the quality of your paper and some typos that need to be edited.

1-I suggest the Authors to add / mention the different types of vaccines that were used in Italy. Indeed, different kind of vaccines were used in different countries (doi: 10.3390/vaccines10050656).

2-Did the Authors know if the students performed different vaccinations (such as for seasonal flu vaccine, or other), as previously reported (doi: 10.3390/vaccines9090943)? Such an information should be fine in order to better underline their behaviour toward general vaccination and sars-CoV2 vaccination.

3- Table 2: the word "frequency" should be in one line.

4- Table 3: the word "frequency" should not be in bold.

5- Table 4: Please, add a sentence explaining that the p-value is statistical significant when written in italics.

6- The word "Sars-CoV2", sometimes is written as "Sars CoV2", other as "Sars-Cov2", other as "SarsCoV2". Please, uniform all over the paper.

7- In the "non-published materials" the questions that allow to answer with numbers, the number 10 is written as "110". Please fix it.

Author Response

Thank you very much for the constructive comments and suggestions. We followed your precious guidance to improve our paper. In attachment we provide a list of changes, point by point.

This manuscript is a resubmission of an earlier submission. The following is a list of the peer review reports and author responses from that submission.

Round 1

Reviewer 1 Report

I do not understand how the sample was drawn. Zje abstract says it was a  convenience sample, whereas the paper itself describes a random sampling of students in a particular uiversity. In either case it is necessary to comparre the socio-economic structeres of the sample with the structures of the universe (the university).

Even in the case of representativeness of the sample for the university, the question of external validity arises. What does 230  undergraduates of one uiniversity can tell us about a national vaccination strategy?

Reviewer 2 Report

The objectives of this study were to obtain information on medical students’ attitudes to ward COVID-19 vaccination and to identify the main barriers to its acceptance. The authors conducted an anonymous online survey on a convenience sample of undergraduate medical students from one  Italian University. The questions were aimed at exploring their attitudes toward vaccination to prevent COVID-19, their perceptions of the risk/threat of COVID-19 and the factors associated with their attitudes toward COVID-19 vaccination.

It is not clear to me what a convenient sample means: please explain

Please make more clear what The Likert-type questions represent

Medical University students are at high risk of COVID-19 infection and/or transmitting the infection to vulnerable people outside the university setting i.e in the hospital during their clinical training such as those with cardiovascular disease  or cancer and this needs to be emphasized

The authors conclude that In the awareness that mandatory measures must be accompanied by initiatives aimed at increasing adherence to vaccination, the present study  demonstrates that the students surveyed are convinced of the importance of vaccination.

I suggest that the authors include as an appendix the online questionnaire translated into English. This would make it more easy for the reader to understand the results of the study.

I think that the study is interesting and would be suitable for publication after minor revisions

Yours sincerely
